# The Oncosuppressive Properties of KCTD1: Its Role in Cell Growth and Mobility

**DOI:** 10.3390/biology12030481

**Published:** 2023-03-21

**Authors:** Giovanni Smaldone, Giovanni Pecoraro, Katia Pane, Monica Franzese, Alessia Ruggiero, Luigi Vitagliano, Marco Salvatore

**Affiliations:** 1IRCCS SYNLAB SDN, Via E. Gianturco 113, 80143 Naples, Italy; 2Institute of Biostructures and Bioimaging, C.N.R., 80134 Napoli, Italy

**Keywords:** KCTD1/KCTD15, colon cancer, WNT/β-catenin signalling, onco-suppressor, biomarker

## Abstract

**Simple Summary:**

Members of the KCTD protein family play major roles in numerous physiopathological functions. Although they are traditionally considered to be involved in neurological and neurodevelopmental processes, there is increasing evidence of their roles as either oncogenic or oncosuppressor factors. Here, we show that KCTD1 has an active role in modulating the stemness and mobility of colon cancer cell lines by affecting the WNT/β-catenin signalling pathway. These findings, which are also supported by analyses of The Cancer Genome Atlas (TCGA) database, provide evidence of the role of KCTD1 as an oncosuppressor.

**Abstract:**

The KCTD protein family is traditionally regarded as proteins that play key roles in neurological physiopathology. However, new studies are increasingly demonstrating their involvement in many other biological processes, including cancers. This is particularly evident for KCTD proteins not involved in protein ubiquitination and degradation, such as KCTD1. We explored the role of KCTD1 in colorectal cancer by knocking down this protein in the human colon adenocarcinoma cell line, SW480. We re-assessed its ability to downregulate β-catenin, a central actor in the WNT/β-catenin signalling pathway. Interestingly, opposite effects are observed when the protein is upregulated in CACO2 colorectal cancer cells. Moreover, interrogation of the TCGA database indicates that KCTD1 downregulation is associated with β-catenin overexpression in colorectal cancer patients. Indeed, knocking down KCTD1 in SW480 cells led to a significant increase in their motility and stemness, two important tumorigenesis traits, suggesting an oncosuppressor role for KCTD1. It is worth noting that similar effects are induced on colorectal cancer cells by the misregulation of KCTD12, a protein that is distantly related to KCTD1. The presented results further expand the spectrum of KCTD1 involvement in apparently unrelated physiopathological processes. The similar effects produced on colorectal cancer cell lines by KCTD1 and KCTD12 suggest novel, previously unreported analogous activities among members of the KCTD protein family.

## 1. Introduction

The KCTD (Potassium Channel Tetramerization Domain) family is an emerging class of proteins involved in a multitude of diverse physiopathological processes [1,2,3,4,5,6,7,8]. The defining feature of this family is the presence of a conserved oligomerization domain denoted as BTB (Broad complex, Tramtrak, and Bric-a-brac) or POZ domain located in the N-terminal region of all its members. On the other hand, sequences of the C-terminal regions appear to share significant similarities only among members belonging to the same clade of the evolutionary tree [9,10,11]. On this basis, the C-terminal regions of these proteins are believed to play a major role in conferring specific functional properties to the KCTD proteins.

However, a full description of the structure–function relationship for these proteins is lacking since structural characterization of the entire protein has been conducted for only two members of the family. Somehow surprisingly, by exploiting the impressive ability of the AlphaFold algorithm [12] to predict protein structures from their sequences, we have shown that most of the C-terminal domains of these proteins share significant structural analogy, thus suggesting closer overall similarities among members of the family [13]. 

Despite these advances in structural characterization, the functional versatility of these proteins that emerges when analysing the literature is puzzling and rather difficult to understand. KCTD proteins were previously implicated in neurological physiopathological processes [1]. However, their implication in other pathologies, including genetic diseases and cancer [2,14,15], is also supported by several studies. Interestingly, the ability of KCTDs to function in diverse biochemical and biological processes is not only evident when these proteins are considered in toto, but is also clear for individual members of the family. In this context, the case of KCTD1 and its close homologue, KCTD15, is illuminating. These proteins were initially implicated in neurodevelopmental disorders [1]. KCTD1 deficiency was then shown to be associated with scalp-ear-nipple syndrome [14,16], whereas KCTD15 was linked with obesity [17]. These two proteins have also been linked with cancers. KCTD15 was recently shown to be overexpressed in human childhood leukaemia [15] and breast cancer [18]. On the other hand, it has been suggested that KCTD1 may act as an oncosuppressor due to its ability to downregulate β-catenin, a central actor in the WNT/β-catenin signalling pathway [2,19,20]. However, the main source of this finding was recently retracted. 

The Wnt/β-catenin pathway plays a key role in embryonic development and tissue homeostasis in adults. Disruption of this pathway often leads to serious diseases such as cancer [21]. The canonical Wnt signalling pathway is regulated by many factors that affect signal transduction in the Wnt/β-catenin signalling pathway [21,22]. It is therefore crucial to identify new and effective regulators of the WNT/β-catenin pathway in order to develop increasingly effective diagnostic and/or therapeutic strategies.

Here, we re-assessed the role of KCTD1 in the WNT/β-catenin pathway by using CRISPR/CAS9 technology to knockdown the protein in the human colon adenocarcinoma cell line, SW480. We also demonstrated that the downregulation of KCTD1 had a major impact on cell motility and stemness. The interplay between KCTD1 and β-catenin was also demonstrated by overexpressing KCTD1 in CACO2 colorectal cancer cells. A potential link between the ability of KCTD1 to downregulate the WNT/β-catenin pathway in these cells and colorectal cancer progression was investigated by interrogating The Cancer Genome Atlas (TCGA) database. Collectively, these data provide support for the role of KCTD1 as an oncosuppressor.

## 2. Materials and Methods

### 2.1. Cell Culture

The human colon adenocarcinoma cell lines SW480 and CACO2 were obtained from the SYNLAB SDN Biobank [23]. The SW480 cell line was grown in RPMI-1640 supplemented with 10% foetal bovine serum (FBS); the CACO2 cell line was grown in EMEM supplemented with 20% FBS. We used CRISPR/CAS9 technology to generate KCTD1-silenced SW480 cells. Briefly, SW480 cells were transfected with the KCTD1 double nickase plasmid (sc-406758-NIC-2, Santa Cruz Biotechnology, Inc., Dallas, TX, USA) to obtain KCTD1-silenced clones (SW480^KCTD1−^). To obtain SW480 control cells (SW480^ctrl^), SW480 cells were transfected with a CRISPR/CAS9 control plasmid (sc-418922, Santa Cruz Biotechnology, Inc., Dallas, TX, USA) plus pReceiver-M94 plasmid (GeneCopoeia, Inc., Rockville, MD, USA). Lipofectamine 3000 reagent (L3000001, Thermo Fisher Scientific, Waltham, MA, USA) was used for the transfection experiments following the manufacturer’s instructions. Puromycin antibiotic (1.5 µg/mL) was added to the cell culture medium for clone selection. All cell lines were cultured at 37 °C in a humidified atmosphere containing 5% CO_2_.

CACO2 and SW480 cells overexpressing KCTD1 (CACO2^KCTD1+^ and SW480^KCTD1+^) were obtained by transiently transfecting these cell lines with a pCMV6-FLAG-KCTD1 expression plasmid (Origene, Rockville, MD 20850, USA). Briefly, CACO2 cells were seeded in a 6-well plate, grown to 70% confluency, and then transfected using Lipofectamine 3000 according to the manufacturer’s instructions. Empty pCMV6 vector was used as a negative control. Cells were harvested 24 h post-transfection, washed with PBS, and then processed for further experiments. 

### 2.2. RNA Extraction and RT-PCR Analysis

Total RNA was extracted from SW480^ctrl^ and SW480^KCTD1−^ cells using the Trizol Reagent protocol (Thermo Fischer Scientific, Waltham, MA, USA). After extraction, RNA was quantified using a spectrophotometer (NP80, Implen, Westlake Village, CA, USA). cDNA was obtained from 0.5 µg of total RNA from each sample, which was reverted using SuperScript III First-Strand Synthesis SuperMix kit (Thermo Fisher Scientific) following the manufacturer’s instructions. qRT-PCR C1000 Touch Thermal Cycler (Bio-Rad, Hercules, CA, USA) was used to detect the expression levels of cyclins using iQ SYBR Green Supermix (#1708882, Bio-Rad). mRNA expression levels of cyclins were quantified using the 2^−∆Ct^ formula. Ribosomal protein S18 (RPS18) was used as an endogenous control to normalize the expression of cyclins. The oligonucleotides used for the qRT-PCR are reported in Table 1. 

### 2.3. Cytofluorimetric Analyses

Flow cytometry experiments were conducted using a minimum of 10,000 recorded events using the Cytoflex V2-B4-R2 instrument (Beckman-Coulter, Brea, CA, USA). The following monoclonal antibodies were used: CD66b-APC (B15091, Beckman Coulter), CD44-PE (A32537, Beckman Coulter), CD90-PC5 (IM3703, Beckman Coulter), and CD133-FITC (W6B3C1, Beckman Coulter). To determine cell cycle progression, the SW480^ctrl^ and SW480^KCTD1^ cells were analysed after 72 h of active growth and controlled by flow cytometry (FCM). For this aim, cells were fixed and stained with propidium iodide (Beckman-Coulter, Brea, CA, USA) and a minimum of 10,000 single-cell events were recorded using the CXP software (Beckman-Coulter, Brea, CA, USA). Then, G0/G1, S, and G2–M phases were determined using the Kaluza Analysis Software 2.1 (Beckman Coulter) and the Michael Fox algorithm.

### 2.4. Vitality and Migration Assays

Cell viability was assessed using the MTT (3-(4,5-dimethylthiazol-2-yl)-2,5-diphenyltetrazolium bromide) assay (G4000, Promega, Madison, WI, USA) and the ATPlite assay (6016943, Perkin Elmer, Waltham, MA, USA). SW480^ctrl^, SW480^KCTD1−^, CACO^2ctrl^, and CACO2^KCTD1+^ cell lines were seeded in 96-well plates at a density of 8 × 103 cells per well. Absorbance was measured at 490 nm for MTT, while the ATPlite luminescence signal was detected using an automatic plate reader (Victor Nivo, Perkin Elmer, Waltham, MA, USA). MTT and ATPlite assays were conducted in triplicate, with similar results. For the cell migration assay, SW480^ctrl^ and SW480^KCTD1−^ cell lines were seeded into the upper Transwell chambers (BD Biosciences, San Jose, CA, USA). After incubation for 24 h, cells that had invaded through the chamber membrane were fixed with 11% glutaraldehyde solution (Sigma-Aldrich, St. Louis, MO, USA ) for 90 min, stained with crystal violet solution and, after elution, quantified through spectrophotometric measurement of optical density (O.D.) at 550 nm. The same experiment was conducted to visualize the invaded cells using a DAPI fluorophore and MICA microscope (Leica, Wetzlar, Germany).

### 2.5. Immunofluorescence

SW480^ctrl^ and SW480^KCTD1−^ cells were seeded in a 6-well plate and a square coverslip placed on top of the plate. Attachment of the knocked-down clones to the coverslip was achieved by letting the setup stand for 4 h. In detail, the medium was gently discarded and the cells washed once with PBS. After checking that a sufficient number of clones remained attached to the coverslip, 4% paraformaldehyde was added to each well and incubated for 15 min at room temperature. Next, the cells were washed twice with Washing Buffer (4% FBS in PBS) and incubated with Permeabilization Buffer (0.3% Triton X-100 in PBS) for 10 min at room temperature, followed by 2 more washes with Washing Buffer and 1 h incubation with Blocking Buffer (2% FBS, 2% BSA, 0.1% Tween20 in PBS). The cells were subsequently incubated with β-catenin antibody (E-5) (sc-7963, Santa Cruz, CA, USA) for 2 h, washed once with Washing Buffer, and then incubated with labelled secondary antibody (Alexa Fluor^®^ 594 Goat anti-mouse IgG, cat# 405326; BioLegend, San Diego, CA, USA) for 45 min. Finally, the cells were washed twice with PBS, incubated for 10 min with DAPI solution (Invitrogen Waltham, MA, USA), and washed once with PBS. The coverslip was then mounted on a microscope slide with a 90:10 glycerol: PBS mounting medium. Confocal images were obtained using a 63× water immersion objective on the MICA microhub (Leica, Wetzlar, Germany) platform, using a 4.4 s exposure time and automatic illumination/pinhole setting. Single planes were acquired for each ROI.

### 2.6. Western Blot Analyses

Expression levels of proteins implicated in cell cycle regulation were evaluated using western blot analysis. In brief, 50 µg of protein extracts derived from SW480^ctrl^, SW480^KCTD1−^_,_ CACO2^ctrl^, and CACO2^KCTD1+^ cell lines were separated using SDS-PAGE. Subsequently, western blotting was performed to check the levels of protein expression. Antibodies used for protein detection are reported in Table 2. ChemiDoc Imaging System (Bio-Rad, USA) coupled with Image Lab software (Bio-Rad, USA) was used for protein acquisition and protein band quantification. Housekeeping genes were used for normalization. 

### 2.7. ELISA

ELISAs were performed using 200 nM of IKK-β (1885150, Invitrogen, USA) in coating and increasing concentrations of recombinant KCTD1 [16] (from 25 to 400 nM). Anti-KCTD1 antibody was used for detection.

## 3. Results

### 3.1. KCTD1 Knockdown Upregulates β-Catenin in SW480 Cells

The interplay between KCTD1 knockdown and the WNT/β-catenin pathway was investigated using the human colorectal adenocarcinoma cell line, SW480 (https://web.expasy.org/cellosaurus/CVCL_0546 accessed on 1 February 2023), in which the protein is significantly expressed [22]. Using the CRISPR/CAS9 protocol that we recently developed for knocking down the related protein, KCTD15, in breast cancer cells [18] showed that the *KCTD1* gene was downregulated in SW480 cells (see Methods for details) (Figure 1A). As shown in Figure 1B and Appendix A, KCTD1-deficient SW480 cells (SW480^KCTD1−^) have similar viability to the parent cells (SW480^ctrl^). Interestingly, SW480^KCTD1−^ cells exhibit a completely different morphology compared with the control cells. Indeed, SW480^KCTD1−^ cells lose the ability to grow in adhesion and begin to grow in suspension (Figure 1C, see also below). Since the literature on other cell lines has indicated a correlation between KCTD1 expression levels and the WNT/β-catenin pathway, we evaluated the effect of KCTD1 knockdown on β-catenin levels. As shown in Figure 1D,E, KCTD1 knockdown significantly upregulated β-catenin expression, thus confirming a negative impact of KCTD1 on the WNT/β-catenin pathway. This finding is in agreement with that obtained by overexpressing KCTD1 in SW480^KCTD1+^ cells, where it is observed that β-catenin levels significantly decrease as KCTD1 levels increase (Appendix A).

### 3.2. KCTD1 and β-Catenin Expression in Colorectal Cancer

Since the upregulation of β-catenin expression and the consequent activation of the WNT/β-catenin pathway have been correlated to an increase in colorectal cancer [24], we evaluated KCTD1 mRNA expression levels in patients with colon adenocarcinoma (COAD) and rectum adenocarcinoma (READ) using data reported in the Cancer Genome Atlas (TCGA). As shown in Figure 2, comparing KCTD1 levels in COAD and READ patients with those observed in normal tissues showed that the protein is downregulated in these patients. Concomitantly, we observed the reverse in the β-catenin trend, i.e., upregulation in the patients. These findings corroborate the observation of the anticorrelation between KCTD1 and β-catenin expression levels and highlights a potential role for KCTD1 as an oncosuppressor in these types of cancers. 

### 3.3. KCTD1 Downregulation Affects Mobility and Stemness of SW480 Cells

As reported above, we observed a significant variation in SW480 cell morphology following KCTD knockdown. On the basis of this observation, we evaluated the effect of KCTD1 knockdown on cell mobility. As shown in Figure 3A,B, SW480^KCTD1−^ cells had significantly higher mobility than SW480^ctrl^ cells. Since increased cell mobility may be associated with increased cell stemness, this aspect was evaluated by performing comparative cytofluorometer analysis of stemness markers (CD66b and CD133). The plot in Figure 3C clearly shows that SW480^KCTD1−^ cells have increased levels of the stemness marker CD66b. Moreover, the concomitant increase in CD133, CD90, and CD44 levels is further confirmation that KCTD1 knockdown induces a shift toward stemness in a significant proportion of the SW480 cells (Appendix A). When the stemness of these cells is highlighted by plotting CD133 versus CD66B, we observe that the percentage of staminal cells increases from the 1.4% observed in SW480^ctrl^ cells to 18.8% in SW480^KCTD1−^ cells. Collectively, these findings provide clear support for the role of KCTD1 as an oncosuppressor in colorectal cancer. 

### 3.4. Effects of KCTD1 Downregulation on Cell Growth and Cell Cycle

To obtain further information on the effects of KCTD1 knockdown, we performed comparative analyses of SW480^KCTD1−^ and SW480^ctrl^ cell growth. As shown in Figure 4A, SW480^KCTD1−^ showed enhanced growth compared with the control cell line. This observation was corroborated using both the MTT assay (Figure 4B) and the ATPlite assay (Figure 4C). These findings are, again, in line with a potential role of KCTD1 as an oncosuppressor. In addition to the phenotypic effects of KCTD1 knockdown, we also monitored possible alterations in the cell cycle. In this context, we observed a significant reduction in the G1 phase coupled with an increase in the G2 phase in SW480^KCTD1−^ cells compared with control cells (Figure 5A). This observation was confirmed by changes in mRNA expression levels of cyclins. Indeed, we detected a significant reduction in cyclin A levels and simultaneous overexpression of cyclin B (Figure 5B). Further, the expression levels of key proteins involved in cell cycle regulation were assessed by western blot. As shown in Figure 5C,D, while no significant differences were observed in the cell cycle activator CDK2, the concomitant reduction in expression levels of the inhibitors p21 and p27 reinforced the finding associated with reduced G1 phase of the cell cycle in SW480^KCTD1−^ cells [25]. The increase in p18 inhibitor expression levels could be associated with an ability of SW480^KCTD1−^ cells to switch more rapidly to the G2/M phase, resulting in greater hyperproliferation of these cells compared with the control cells. Finally, it is important to note that levels of the well-known apoptosis activator p53 are significantly lower in SW480^KCTD1−^ cells than in control cells, in line with their increased growth rate. 

### 3.5. KCTD1 Knockdown Determines Increased Transcriptional Activity of β-Catenin

To assess the effect of KCTD1 downregulation on β-catenin action, we performed confocal microscopy experiments on SW480 cells. As shown in Figure 6A and Appendix A, knocking down KCTD1 results in a clear displacement of β-catenin in the nuclei of SW480^KCTD1−^ cells compared with the nuclei of control cells. This increased nuclear localisation of β-catenin was also corroborated by evaluating the expression levels of its different targets. As shown in Figure 6B, *ABCB1*, *DKK4*, *LGR5*, *TCF1* and *cMYC* levels are significantly increased in SW480^KCTD1−^ cells, while ENFB1 levels are significantly decreased, as expected [26].

### 3.6. KCTD1 Overexpression Reduces Colon Cancer Cell Growth

To validate and expand the results obtained using SW480 cells, we overexpressed KCTD1 in another colon cancer cell line, CACO2, which expresses lower levels of KCTD1 compared with SW480 cells (data not shown). KCTD1 overexpression resulted in significant reduction in β-catenin levels in CACO2^KCTD1+^ cells compared with control CACO2^ctrl^ cells (Figure 7A,B). KCTD1 overexpression also significantly inhibited CACO2 cell growth (Figure 7C), contrary to the growth observed in SW480^KCTD1−^ cells (see Figure 4C). Furthermore, significant deregulation of different β-catenin target genes was observed, suggesting a reduction in the activity of β-catenin transcription factors (Figure 7D). We observed significant reduction in *ABCB1*, *DKK4*, *LGR5*, *TCF1* and *cMYC* β-catenin target genes in CACO2^KCTD1+^ cells. Finally, the upregulation of KCTD1 resulted in significant reduction in the G2–M phase of the cell cycle, which induced an increase in the G1 phase in CACO2^KCTD1+^ cells compared with CACO2^ctrl^ cells (Figure 7E). The overexpression of KCTD1 was also associated with significant reduction in cyclin E mRNA levels.

## 4. Discussion

Traditionally seen as proteins playing key roles in neurological physiopathology, research activities focused on KCTD proteins are progressively showing their substantial involvement in many other biological processes, including cancers [1,2]. The fact that these proteins play important roles in diverse biological process is indicative of their involvement in basic, yet crucial, biochemical activities. This is not surprising for the KCTD proteins that work as cullin 3 ligase substrate adaptors in protein ubiquitination and degradation [27,28]. Indeed, these KCTDs may cooperate in the regulation of proteins involved in different biological processes. However, defining the roles of KCTDs that are unable to bind cullin 3 and, consequently, to act as ligase substrate adaptors, appears to be quite difficult [9,27,28]. Among these, KCTD1 and its close homolog KCTD15—whose activity has been detected in several unrelated physiopathological processes—are particularly interesting. Moreover, their role in cancer has not been completely explored. 

In this study, we explored the potential role of KCTD1 in colorectal cancer. In particular, by knocking down KCTD1 in the human colon adenocarcinoma cell line, SW480, using CRISPR/CAS9 technology, we re-assessed its ability to downregulate β-catenin, a central actor in the WNT/β-catenin signalling pathway. As alterations in Wnt/β-catenin signalling have been strongly associated with colorectal cancer tumorigenesis, metastasis and recurrence [29], we interrogated the TCGA database to look for KCTD1 and β-catenin misregulation and showed that KCTD1 downregulation was associated with β-catenin overexpression in colorectal cancer patients. β-catenin is localized in the nuclei of KCTD1-deficient SW480 cells and its transcription factor activity is altered in SW480^KCTD1−^ cells. The role of KCTD1 as an oncosuppressor in this cancer was sustained by other important alterations induced by protein knockdown in SW480 cells. In particular, knocking down the protein significantly changed the morphology of the cell, with a significant increase in their motility, in line with the known ability of β-catenin to induce cell migration. Moreover, we also observed a significant increase in their stemness following KCTD1 knockdown, again as the result of the concomitant β-catenin upregulation [30]. As acquired cell motility and stemness represent important traits of tumorigenesis, these findings concur to delineate a role for KCTD1 as an oncosuppressor. It is worth noting that similar effects on colorectal cancer cells are induced by the misregulation of KCTD12, a protein that is distantly related to KCTD1. Indeed, it has been reported that KCTD12 downregulation or overexpression dramatically increases and represses colorectal cancer cell stemness [31]. Since cell cycle dysregulation is a common mechanism for uncontrolled cell proliferation and tumorigenesis, we also evaluated the impact of KCTD1 knockdown on the cell cycle. Our data indicate that knocking down the protein leads to a significant reduction in the G1 phase coupled with an increase in the G2 phase. We also observed an increase in p18 inhibitor levels coupled with a downregulation of the well-known apoptosis activator, p53, in line with increased growth of the SW480^KCTD1−^ cells. The expression levels could be associated with an ability of SW480^KCTD1−^ cells to switch more rapidly to the G2/M phase, resulting in greater hyperproliferation of these cells compared with control cells. Finally, it is important to note that levels of the well-known apoptosis activator p53 are significantly lower in SW480^KCTD1−^ cells than in control cells, in line with their increased growth rate. The ability of KCTD1 to deregulate the action of β-catenin was confirmed in CACO2 cells, another colon adenocarcinoma cell line. The overexpression of KCTD1 in this cell line induced opposite effects to those observed upon KCTD1 downregulated in SW480 cells. Indeed, we observe a downregulation of β-catenin in CACO2^KCTD1+^ cells and a reduction in its ability to transcribe different β-catenin target genes. Furthermore, the upregulation of KCTD1 also resulted in a reduction in the proliferative capacity of CACO2 cells by increasing the G1-phase, which was associated with concomitant reduction in the G2–M-phase. Although the molecular mechanism underlying the effects of KCTD in this cell line require further studies, some working hypotheses can be proposed. It is possible that these effects are mediated by transcription factor AP-2α, a well-characterized KCTD1 interactor. It has recently been shown that AP-2α, independently or together with KCTD1, can downregulate β-catenin, likely by stabilizing the APC–β-catenin complex, thus preventing the transcriptional activity of β-catenin and repressing the WNT/β-catenin pathway. In this scenario, the ability of KCTD1 to bind to AP-2α may further stabilize the complex with consequent downregulation of β-catenin activity. Alternatively, or concomitantly, KCTD1 could function like KCTD15, which has recently been shown to upregulate the phosphorylated active form of IKK-β, which is able to phosphorylate β-catenin, favouring its degradation. This possibility, which would open a new crosstalk between NF-kβ and the WNT/β-catenin pathway, is supported by the ability of recombinant KCTD1 to directly bind recombinant IKK-β in in vitro assays (Appendix A). 

This experimental scenario coupled with the observation here reported of the downregulation and upregulation of KCTD1 and β-catenin expression, respectively, in patients with colon and rectum adenocarcinoma is suggestive of the role of KCTD1 as a potential oncosuppressor in these types of cancers. Interestingly, the evaluation of the expression levels of KCTD12 and KCTD15—which are related to KCTD1—in colorectal cancer using TCGA data highlights similar downregulation (Appendix A). This suggests that different members of the family may play an active role in this pathology.

## 5. Conclusions

The presented results further expand the spectrum of KCTD1 involvement in apparently unrelated physiopathological processes. Although the precise mechanism underlying β-catenin downregulation by KCTD1 has not been fully assessed, the present finding highlights the possibility of exploring new diagnostic and therapeutic strategies for colorectal cancer. Moreover, the similarities in the effects of KCTD1 and KCTD12 on colorectal cancer cell lines suggests novel, previously unreported similarities in the activities of members of the KCTD protein family.

## Figures and Tables

**Figure 1 biology-12-00481-f001:**
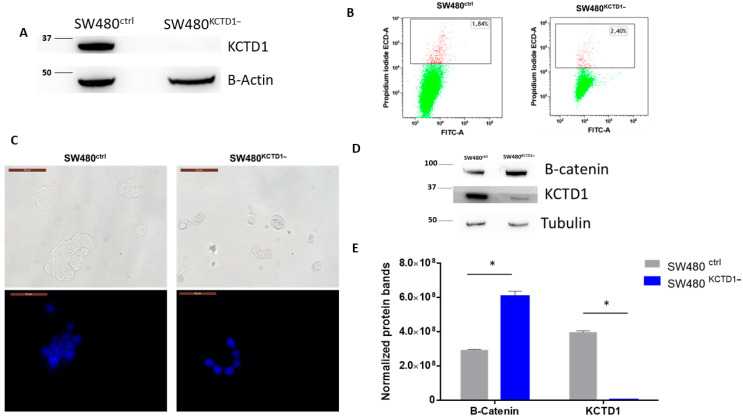
(**A**) Western blot analysis of SW480^ctrl^ and SW480^KCTD1−^ cells. Numbers represent molecular weights expressed in Kda. (**B**) Propidium iodide VS FITC Dot plot analysis of SW480^ctrl^ and SW480^KCTD1−^ cells. Numbers represent the propidium iodide-positive cells. (**C**) Brightfield- (top panels) and DAPI-stained (lower panels) microscopy of SW480^ctrl^ and SW480^KCTD1−^ cells. Scale Bar 50 µm. Magnification 40×. Western blot analysis of the indicated proteins (**D**) and protein band quantifications (**E**) were performed on SW480^ctrl^ (grey bars) and SW480^KCTD1−^ (blue bars) cells after 72 h of active growth. Numbers represent molecular weights of proteins expressed in kDa. * = *p*-value < 0.05.

**Figure 2 biology-12-00481-f002:**
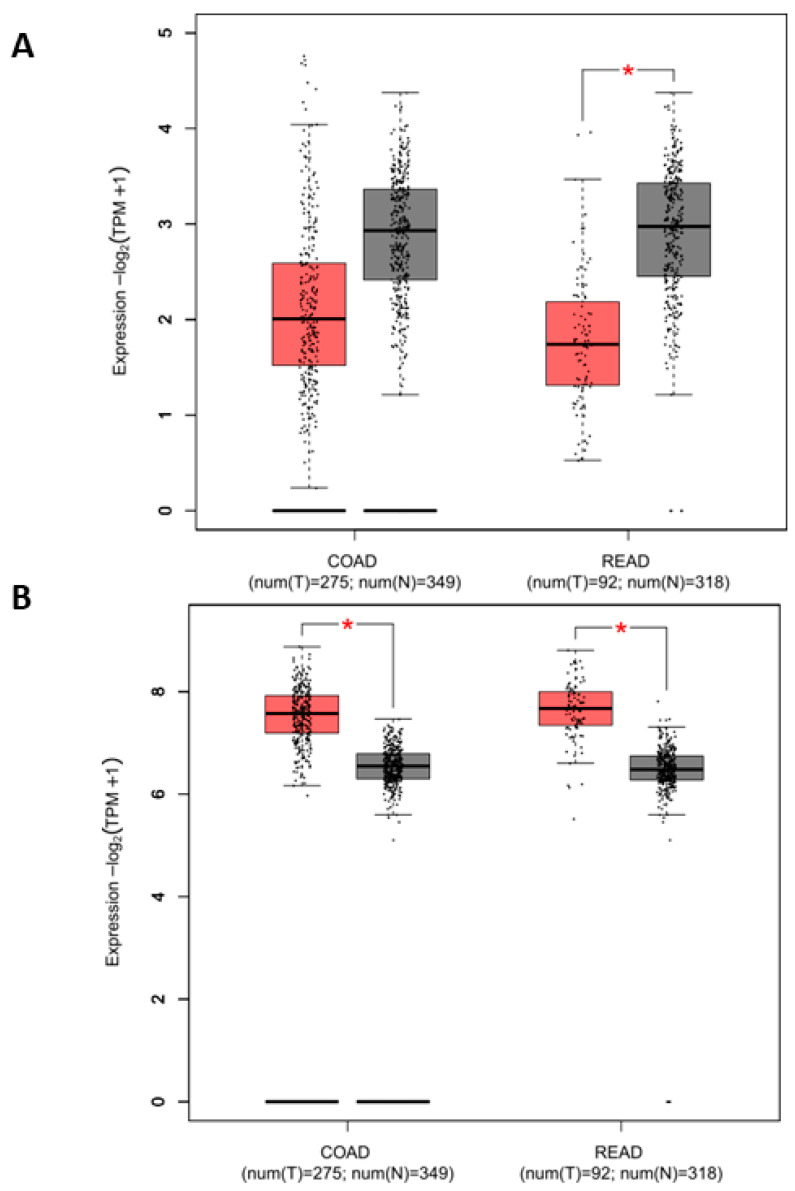
KCTD1 ((**A**), top panel) and β-catenin ((**B**) low panel) mRNA expression in colon adenocarcinoma (COAD) and rectum adenocarcinoma (READ) and normal tissues (GEPIA2). Box plots show median KCTD1 mRNA expression levels in tumour tissues (red plots) and normal tissues (grey plot). Axis units are Log2 (TPM + 1). * red = *p*< 0.01. T = tumour tissues. N = normal tissues.

**Figure 3 biology-12-00481-f003:**
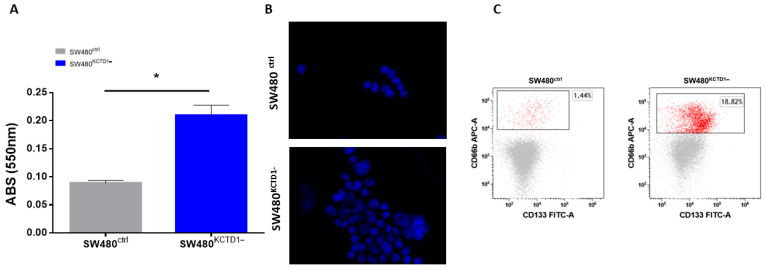
(**A**) SW480^ctrl^ and SW480^KCTD1−^ cells were allowed to invade the Transwell insert for 24 h. Invading cells were stained with crystal violet and the signal recorded by measuring absorbance at 550 nm. Cells were also photographed after staining the nuclei with DAPI (**B**). CD66-APC VS CD133-FITC dot plots of SW480^ctrl^ and SW480^KCTD1−^ cells (**C**). Percentage of gated cells. Bars represent the mean of triplicate experiments ± standard deviations. * = *p* < 0.05.

**Figure 4 biology-12-00481-f004:**
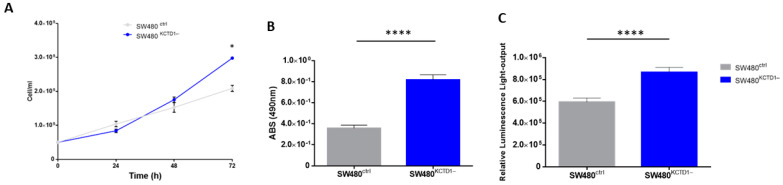
(**A**) SW480^ctrl^ (grey line) and SW480^KCTD1−^ (blue line) cell growth (expressed as cells/µL). Cell concentrations are shown as mean +/− SD of three technical independent experiments. * = *p* < 0.05. MTT assay (**B**) and ATPlite assay (**C**) of SW480^ctrl^ (grey bars) and SW480^KCTD1−^ (blue bars) after 72 h of growth. **** = *p* < 0.001. Mann–Whitney *t*-test. Error bars represent the SD of four independent experiments.

**Figure 5 biology-12-00481-f005:**
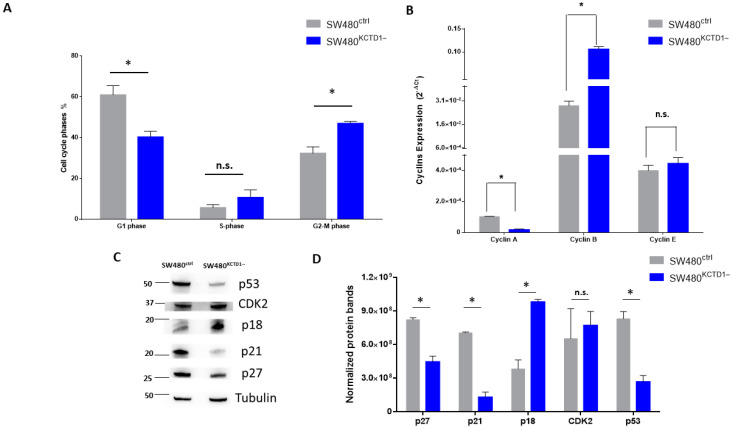
(**A**) The percentages of cells in the G0–G1, S, and G2–M phases after 72 h of active growth were reported as the mean values of three independent experiments +/− SD. * = *p* < 0.05. Mann–Whitney *t*-test (**B**) mRNA expression levels of cyclins in SW480^ctrl^ (grey bars) and SW480^KCTD1−^ (blue bars) after 72 h of active growth. Relative expression was determined using the 2^−ΔCt^ method. Relative expression of cyclins is shown as mean +/− SD of two technical independent experiments. * = *p* < 0.05, Mann–Whitney *t*-test. Western blot analysis of the indicated proteins (**C**) and protein band quantifications (**D**) were performed in SW480^ctrl^ (grey bars) and SW480^KCTD1−^ (blue bars) after 72 h of active growth. Numbers represent molecular weights of proteins expressed in kDa. ns = not significant; * = *p* < 0.05.

**Figure 6 biology-12-00481-f006:**
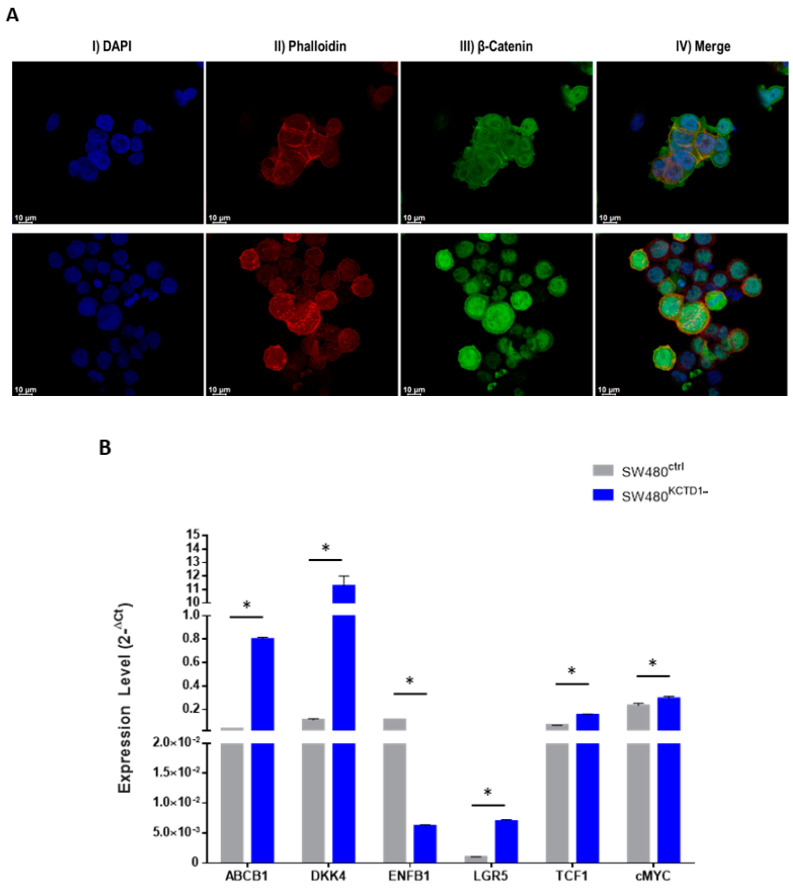
(**A**) Immunofluorescence analysis of SW480^ctrl^ (upper panels) and SW480^KCTD1−^ (lower panels). (I) Nucleus staining with DAPI (blue), and (II) Phalloidin-Dye light 554. (III) β-catenin staining with Alexa588-conjugated secondary antibody. (IV) Merging of DAPI, Dye light 554, and Alexa 488. Magnification 63×. Scale bars 10 µm. (**B**) mRNA expression levels of β-catenin targets in SW480^ctrl^ (grey bars) and SW480^KCTD1−^ (blue bars) cells after 72 h of active growth. Relative expression was determined using the 2^−ΔCt^ method. Relative expressions of β-catenin target genes are shown as mean +/− SD of three technical independent experiments. * = *p* < 0.05, Mann–Whitney *t*-test.

**Figure 7 biology-12-00481-f007:**
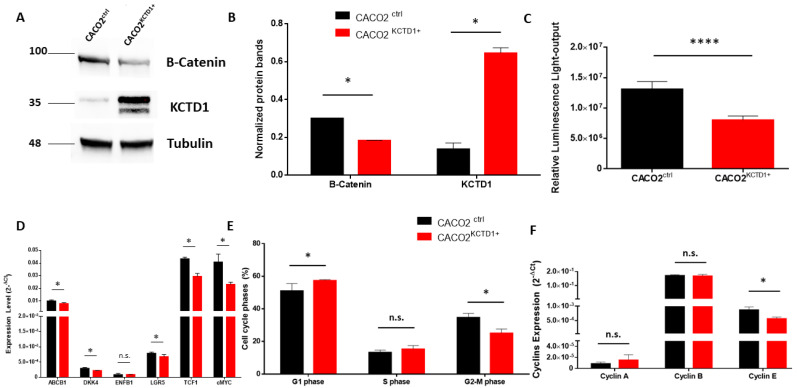
Western blot analysis of the indicated proteins (**A**) and protein band quantifications (**B**) were performed on CACO2^ctrl^ cells (black bars) and CACO2^KCTD1+^ cells (red bars) after 24 h of overexpression. Numbers represent molecular weights of proteins, expressed in kDa. * = *p* < 0.05. (**C**) ATPlite assay of CACO2^ctrl^ cells (black bars) and CACO2^KCTD1+^ cells (red bars) after 24 h of overexpression. **** = *p* < 0.001. Mann–Whitney *t*-test. Error bars represent the SD of four independent experiments. (**D**) β-catenin targets mRNA expression levels in CACO2^ctrl^ cells (black bars) and CACO2^KCTD1+^ cells (red bars). Relative expression was determined using the 2^−ΔCt^ method. Relative mRNA expressions of β-catenin targets are shown as mean +/− SD of three technical independent experiments. * = *p* < 0.05, Mann–Whitney *t*-test. (**E**) The percentage of CACO2^ctrl^ cells (black bars) and CACO2^KCTD1+^ cells (red bars) in the G0–G1, S, and G2–M phases after 24 h of KCTD1 overexpression were reported as the means of three independent experiments +/− SD * = *p* < 0.05. Mann–Whitney *t*-test. (**F**) mRNA expression levels of cyclins in CACO2^ctrl^ cells (black bars) and CACO2^KCTD1+^ cells (red bars). Relative expression was determined using the 2^−ΔCt^ method. Relative cyclin expression is shown as mean +/− SD of three technical independent experiments. ns = not significant; * = *p* < 0.05, Mann–Whitney *t*-test.

**Table 1 biology-12-00481-t001:** Oligonucleotide sequences used for qRT-PCR.

Gene Name	Oligo Forward	Oligo Reverse
RPS18	5′-CGATGGGCGGCGGAAAATA-3′	5-CTGCTTTCCTCAACACCACA-3′
CyclinA	5′-AAATGGGCAGTACAGGAGGA-3′	5′-CCACAGTCAGGGAGTGCTTT-3′
CyclinB	5′-CATGGTGCACTTTCCTCCTT-3′	5′ AGGTAATGTTGTAGAGTTGGTGTCC-3′
CyclinE	5′-GGCCAAAATCGACAGGAC-3′	5′-GGGTCTGCACAGACTGCAT-3′

**Table 2 biology-12-00481-t002:** Antibodies used for western blot analysis.

Protein Name	Antibody
KCTD1	PA5-24877, ThermoFisher
p27 Kip1	D69C12, Cell Signaling Technology (Danvers, MA, USA) (Cat# 13715)
p21 Waf1/Cip1	Cell Signaling Technology Cat# 2947
CDK2	78B2, Cell Signaling Technology, Cat #2546
FLAG peptide	MA1-91878, Thermo Fisher
β-catenin	MA1-301, Thermo Fisher Sc-7963, SantaCruz
β-Actin	MA1-140, Thermo Fisher
β-tubulin	Sigma-Aldrich Cat# T0198

## Data Availability

The data used and analysed in this study are available from the corresponding author on reasonable request.

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
