# Peer review of "The Oncosuppressive Properties of KCTD1: Its Role in Cell Growth and Mobility"

_biology, 2023, doi:10.3390/biology12030481_

Round 1

Reviewer 1 Report

The article by Smaldone et. al. on the oncosuppressive properties of KCTD1 is very interesting and a good study on the novel role of this protein in cancer.  However, there are three major concerns with this study and a few additional minor issues.

Major Concerns

1.       The knockdown of KCTD1 and the impact must be confirmed in additional colorectal cancer cell lines, at least 2 (1 with an APC mutation similar to SW480, such as HT29 or DLD1 cells and a second cell line with a B-catenin mutation such as HCT116 or LS174).  The use of 1 cell line to show an effect is not sufficient.

2.       A follow-up study in SW480 cells that shows that overexpression of KCTD1, reduces the levels of B-catenin, or otherwise adversely impacts cell growth is needed.

3.       The impact of KCTD1 on Wnt/B-catenin impact should be addressed, either through the examination of target gene expression (myc, cyclin-d1, survivin, etc) and/or TOP/FOP flash reporter assays.

 Minor Concerns

1.       In the Simple Summary, the phrase “Recently their role in cancer regulation” is repeated in the 2nd and 3rd sentences.  These sentences should be combined.

2.       The antibody for CDK2 in Table 2 is a primer sequence not an antibody

3.       The formatting of reference citations is not consistent throughout the manuscript.

Overall, this study is interesting, but is missing key confirmatory experiments to show the importance of KCTD1 in colorectal cancer cell growth.

Author Response

Comments and Suggestions for Authors

The article by Smaldone et. al. on the oncosuppressive properties of KCTD1 is very interesting and a good study on the novel role of this protein in cancer.  However, there are three major concerns with this study and a few additional minor issues.

Overall, this study is interesting, but is missing key confirmatory experiments to show the importance of KCTD1 in colorectal cancer cell growth.

Response

We thank the reviewer for the positive evaluation of the manuscript and for the constructive criticisms that have been addressed, also performing new experiments, as detailed below.

Major Concerns

  1. The knockdown of KCTD1 and the impact must be confirmed in additional colorectal cancer cell lines, at least 2 (1 with an APC mutation similar to SW480, such as HT29 or DLD1 cells and a second cell line with a B-catenin mutation such as HCT116 or LS174).  The use of 1 cell line to show an effect is not sufficient.
  2. A follow-up study in SW480 cells that shows that overexpression of KCTD1, reduces the levels of B-catenin, or otherwise adversely impacts cell growth is needed.
  3. The impact of KCTD1 on Wnt/B-catenin impact should be addressed, either through the examination of target gene expression (myc, cyclin-d1, survivin, etc) and/or TOP/FOP flash reporter assays.

Response

We thank the reviewer for raising these important issues that we addressed by performing new experiments that support and expand the conclusions of the original manuscript. The need to further validate our conclusions by using additional cell lines (point #1) and by overexpressing the protein in SW480 cells (point #2) was here addressed by considering the CACO2 colorectal cancer cell line in which KCTD1 is minimally expressed. Instead of knocking-down the protein, the low amount of KCTD1 endogenously present in this cell line gave us the opportunity to evaluate the effect of KCTD1 over-expression. Interestingly, the upregulation of KCTD1 reversed the effects induced by its upregulation in SW480 cells. Indeed, we observed a reduction in cell growth and in beta-catenin levels. As requested by the reviewer, we also evaluated the effect of KCTD1 up and downregulation on c-Myc (point #3). Again, up and downregulation of the protein had opposite effects on c-Myc mRNA levels. The data emerged from these experiments fully corroborates the conclusions reported in the original manuscript. Moreover, upon revision, we also evaluated the role of KCTD1 in affecting the localization of beta-catenin in SW480 cells.

Minor Concerns

  1. In the Simple Summary, the phrase “Recently their role in cancer regulation” is repeated in the 2ndand 3rd  These sentences should be combined.

Response

This typo has been corrected in the revised version. We thank the reviewer for pointing it out.

The antibody for CDK2 in Table 2 is a primer sequence not an antibody

Response

This error has been corrected in the revised version. We thank the reviewer for pointing it out.

  1. The formatting of reference citations is not consistent throughout the manuscript.

Response

The style of the references has been uniformed throughout the text

Reviewer 2 Report

Authors identified KCTD1 as an onco-suppressor in human colon adenocarcinoma SW480 cell line, using CRISPR/CAS9 technology. Consistent with colorectal cancer patient data from TCGA database, β-catenin expression level is negatively associated to KCTD1 level. Then authors characterized silencing KCTD1 could promote cell motility and stemness, two traits of tumorigenesis, suggesting KCTD1 likely functions through WNT/B-catenin to suppress tumor progression. Moreover, authors analyzed cell cycle progression and tested cell cycle regulators that might be involved. Interestingly, p53, a well-known tumor suppressor which also act as a crucial cell cycle regulator, was down regulated in KCTD1 silenced cells. This study provides insight into KCTD1’s novel function as an onco-suppressor, and also has significant clinical potentials.

Major points:

1.     I’m confused about how the cell line SW480KCTD1 was generated. In abstract (line 21) and many other places in the text it’s described as “silencing” or “knocking-down” (line 77) but it’s also described as “knock-out” (line 81). In my understanding “silencing” means negative regulation of expression. In figure 1D there was a week band of KCTD1 which suggests it’s “silencing”. Could authors elaborate more in detail about the approach to silence KCTD1 using CRISPR/CAS9 which mostly used to introduce “knock-out” but in some cases also can inactivate gene expression. And indicates explicitly if there are two different strategies knock-down vs. knock-out that were used in the study. 

2.     It’s a bit concerning that the major point of the study is similar to a retracted paper (Xinxin Li et al., 2014. doi: 10.1371/journal.pone.0094343), as authors mentioned as well (line 68-69). Indeed, the retraction was in light of the concerns of figure 4C and 5B of the retraction paper which were not relevant to β-catenin. Though figure 7 Effects of KCTD1 on the expression of β-catenin protein and 8C “Effects of KCTD1 on β-catenin ubiquitination and the expression of Wnt/β-catenin downstream genes and AP-2α.” also have image manipulation, authors managed to clarify their mistakes and editors accepted (The PLOS ONE Editors, 2022. doi: 10.1371/journal.pone.0268604). In terms of research novelty, in their second paper “KCTD1 inhibited β‑catenin expression levels” (not retracted), used same cell line and performed similar experiments, which could still impact the novelty of this study. (Lingyu Hu, et al., 2020. DOI: 10.3892/mmr.2020.11457)

 Authors mentioned KCTD12 in abstract and listed KCTD15 in keywords. KCTD12 is distant-related to KCTD1 but misregulated in colorectal cancer cells while KCTD15 is close-related but not reported in physio-pathological processes. It would make sense to test if they share similar mechanisms via β-catenin in tumorigenesis. A simple way is to perform a same analysis of TCGA database like figure 2.

Minor points:

Figures:

Titles in different panels have different fonts, sizes and formats. Many panels not aligned.

1A, needs to be consistent, with or without “Anti” when indicating protein bands.

1B, gatings (box that includes positive cells) are different in the two plots, also in 3C. Please indicates replicates, or SD or p-value for FACS analysis.

1C, brightfiled and fluorescent images don’t look corresponding.

1D, KCTD1 band has two different background colors.

1E, Some strange frames inside the blue bar. P-values especially the one on the right looks like <0.001 to me, and also in figure 3A.

3A, ABS 490nm in the plot but 550nm in figure legend.

3B, please move titles to the left.

Many typos, for example:

line 11, “Recently their role in cancer regulation.” Incomplete sentence.

line 26, “trait” and line 57, “is” should be plural.

line 66, “may play act as an oncosuppressor”

line 72 and line 102, “witch”

line 76, spaces missing in the first line of the paragraph

line 90, “obtaine”

line 121, “10.000 single-cell events” should be “10,000 single-cell events”

line 158, "KCTD1 gene", gene name italic

“SW480KCTD1” in section2.3 and section2.4 should be “SW480KCTD1

References in section 3.1 and 3.2 missing “[ ]”

Author Response

Reviewer 2

English language and style

( ) English very difficult to understand/incomprehensible
(x) Extensive editing of English language and style required
( ) Moderate English changes required
( ) English language and style are fine/minor spell check required
( ) I don't feel qualified to judge about the English language and style

Yes

Can be improved

               Must be improved

Does the introduction provide sufficient background and include all relevant references?

( )

(x)

( )

( )

Are all the cited references relevant to the research?

(x)

( )

( )

( )

Is the research design appropriate?

(x)

( )

( )

( )

Are the methods adequately described?

( )

( )

(x)

( )

Are the results clearly presented?

(x)

( )

( )

( )

Are the conclusions supported by the results?

(x)

( )

( )

( )

Comments and Suggestions for Authors

Authors identified KCTD1 as an onco-suppressor in human colon adenocarcinoma SW480 cell line, using CRISPR/CAS9 technology. Consistent with colorectal cancer patient data from TCGA database, β-catenin expression level is negatively associated to KCTD1 level. Then authors characterized silencing KCTD1 could promote cell motility and stemness, two traits of tumorigenesis, suggesting KCTD1 likely functions through WNT/B-catenin to suppress tumor progression. Moreover, authors analyzed cell cycle progression and tested cell cycle regulators that might be involved. Interestingly, p53, a well-known tumor suppressor which also act as a crucial cell cycle regulator, was down regulated in KCTD1 silenced cells. This study provides insight into KCTD1’s novel function as an onco-suppressor, and also has significant clinical potentials.

Response

We thank the reviewer for the positive evaluation of the manuscript and for the constructive criticisms that have been addressed, also performing new experiments, as detailed below.

Major points:

  1. I’m confused about how the cell line SW480KCTD1 was generated. In abstract (line 21) and many other places in the text it’s described as “silencing” or “knocking-down” (line 77) but it’s also described as “knock-out” (line 81). In my understanding “silencing” means negative regulation of expression. In figure 1D there was a week band of KCTD1 which suggests it’s “silencing”. Could authors elaborate more in detail about the approach to silence KCTD1 using CRISPR/CAS9 which mostly used to introduce “knock-out” but in some cases also can inactivate gene expression. And indicates explicitly if there are two different strategies knock-down vs. knock-out that were used in the study. 

Response

We thank the reviewer for highlighting this important aspect. Indeed, this comment offers us the opportunity to make an important clarification. The CRISPR/CAS9 approach was used to knock-out KCTD1 in SW480 cell line. Although in treated cells the expression of KCTD1 was greatly reduced, a marginal but detectable amount of the protein was still detected. Incomplete editing events in cancer cell lines may occur due to the variability of the number of gene copy based on genomic heterogeneity (see doi: 10.3892/ol.2017.7605). In this regard, it is important to note that SW480 cells are hyper-diploid (doi:10.1016/j.cancergencyto.2004.03.014). In this scenario, we removed the expression knock-out and replace it with 'knock-down' upon revision. Moreover, to avoid confusion, we also eliminated the term silencing. The incomplete gene editing produced by our CRISP/Cas9 experiments has been mentioned in the revised manuscript.

  1. It’s a bit concerning that the major point of the study is similar to a retracted paper (Xinxin Li et al., 2014. doi: 10.1371/journal.pone.0094343), as authors mentioned as well (line 68-69). Indeed, the retraction was in light of the concerns of figure 4C and 5B of the retraction paper which were not relevant to β-catenin. Though figure 7 “Effects of KCTD1 on the expression of β-catenin protein”and 8C “Effects of KCTD1 on β-catenin ubiquitination and the expression of Wnt/β-catenin downstream genes and AP-2α.” also have image manipulation, authors managed to clarify their mistakes and editors accepted (The PLOS ONE Editors, 2022. doi: 10.1371/journal.pone.0268604). In terms of research novelty, in their second paper “KCTD1 inhibited β‑catenin expression levels” (not retracted), used same cell line and performed similar experiments, which could still impact the novelty of this study. (Lingyu Hu, et al., 2020. DOI: 10.3892/mmr.2020.11457).

Response.

Although, as correctly noticed by the reviewer, there is some overlap of the present manuscript with the PlosOne retracted paper and with the more recent study based on the functional analysis of KCTD1 mutants related the SEN syndrome, the present manuscript presents novel intriguing indications on the role of KCTD1 in cell cycle, cell migration and possibly in colon cancer. (DOI: 10.3892/mmr.2020.11457)

  1. Authors mentioned KCTD12 in abstract and listed KCTD15 in keywords. KCTD12 is distant-related to KCTD1 but misregulated in colorectal cancer cells while KCTD15 is close-related but not reported in physio-pathological processes. It would make sense to test if they share similar mechanisms via β-catenin in tumorigenesis. A simple way is to perform a same analysis of TCGA database like figure 2.

Response.

We thank the reviewer for suggesting this interesting analysis. Upon revision, we evaluated the expression of KCTD12 and KCTD15 in colorectal cancer using TCGA data. As can be seen from newly added Supplementary Figure 5 of the revised manuscript, the levels of KCTD12 and KCTD15 in colorectal cancer follow those of KCTD1, suggesting that they may play a similar functional role in this pathology. This finding has been commented in the Discussion section

Response.

Although, as correctly noticed by the reviewer, there is some overlap of the present manuscript with the PlosOne

Minor points:

Figures:

Titles in different panels have different fonts, sizes and formats. Many panels not aligned.

1A, needs to be consistent, with or without “Anti” when indicating protein bands.

Response.

Figure 1A has been modified considering the suggestions of the reviewer.

1B, gatings (box that includes positive cells) are different in the two plots, also in 3C. Please indicates replicates, or SD or p-value for FACS analysis.

Response

We added the replicates of FACS analysis in the Supplementary Figures of the revised version of the manuscript for the Figure 1B and the SD for the Figure 3C.

1C, brightfiled and fluorescent images don’t look corresponding.

Response

Brightfield and fluorescent images do not represent the same field

1D, KCTD1 band has two different background colors.

Response

Please refer to the “Uncropped WB images” file to see that there are no modifications in figure 1D.

1E, Some strange frames inside the blue bar. P-values especially the one on the right looks like <0.001 to me, and also in figure 3A.

Response

Due to the small number of independent experiments (triplicates), the T-Test performed on these data only provides a p-value<0.05.

3A, ABS 490nm in the plot but 550nm in figure legend.

Response

We modified the revised version of the manuscript. We apologize for the error and we thank the reviewer for pointing it out.

3B, please move titles to the left.

Response

Done.

Many typos, for example:

line 11, “Recently their role in cancer regulation.” Incomplete sentence.

line 26, “trait” and line 57, “is” should be plural.

line 66, “may play act as an oncosuppressor”

line 72 and line 102, “witch”

line 76, spaces missing in the first line of the paragraph

line 90, “obtaine”

line 121, “10.000 single-cell events” should be “10,000 single-cell events”

line 158, "KCTD1 gene", gene name italic

 “SW480KCTD1” in section2.3 and section2.4 should be “SW480KCTD1

References in section 3.1 and 3.2 missing “[ ]”

Response

All these typos have been corrected.

Round 2

Reviewer 1 Report

The revised article by Smaldone et. al. on the oncosuppressive properties of KCTD1 addresses some of my original concerns but does not adequately address them all. 

Major Concerns

1.       The knockdown of KCTD1 and the impact must be confirmed in additional colorectal cancer cell lines, a third additional cell line is still needed to confirm their results in particular one with a B-catenin mutation such as HCT116 or LS174.  The use of 2 cell lines with similar mutations is not sufficient. Nor does it fully address my original concern, particularly since they two cell lines were used in different experiments.

2.       The data showing β-catenin localization and gene activation in SW480KCTD1- cells is flawed, and the presented data do not support the findings reported by the authors. In particular, the authors claim there is less nuclear β-catenin in the knockout cells, which does not appear to be easily observed in the images presented in Figure 6A.  Furthermore, they show an increase (albeit very small) in a β-catenin target gene Myc, which argues against a decrease in nuclear levels of β-catenin in Figure 6B.  As noted with the cell lines above, one gene is not sufficient to say that gene expression by a transcription factor is up or down regulated.  Additional, genes would need to be examined.     

3.       The follow-up study in CACO2 cells is interesting, but does not fully address the concern I originally presented.  The same cell line should be used for both a knockdown and overexpression study.  Therefore, the authors need to show knockdown of KCTD1 in CACO2 has the opposite effect as overexpression or that overexpression in SW480 cells decreases cell growth. 

Overall, this study is interesting, but is missing key confirmatory experiments to show the importance of KCTD1 in colorectal cancer cell growth.

Author Response

Reviewer 1

English language and style

( ) English very difficult to understand/incomprehensible
( ) Extensive editing of English language and style required
( ) Moderate English changes required
(x) English language and style are fine/minor spell check required
( ) I don't feel qualified to judge about the English language and style

Yes

Can be improved

Must be improved

Not applicable

Does the introduction provide sufficient background and include all relevant references?

(x)

( )

( )

( )

Are all the cited references relevant to the research?

(x)

( )

( )

( )

Is the research design appropriate?

( )

( )

(x)

( )

Are the methods adequately described?

( )

(x)

( )

( )

Are the results clearly presented?

( )

(x)

( )

( )

Are the conclusions supported by the results?

( )

(x)

( )

( )

Comments and Suggestions for Authors

The revised article by Smaldone et. al. on the oncosuppressive properties of KCTD1 addresses some of my original concerns but does not adequately address them all. 

Major Concerns

  1. The knockdown of KCTD1 and the impact must be confirmed in additional colorectal cancer cell lines, a third additional cell line is still needed to confirm their results in particular one with a B-catenin mutation such as HCT116 or LS174.  The use of 2 cell lines with similar mutations is not sufficient. Nor does it fully address my original concern, particularly since they two cell lines were used in different experiments.

Response: As we do not own the cell lines suggested by the reviewer available, we found it useful to over-express KCTD1 in both a colon cancer cell system with low levels of the KCTD1 protein and in SW480. The results of these new experiments are reported in the revised version of the manuscript.

  1. The data showing β-catenin localization and gene activation in SW480KCTD1-cells is flawed, and the presented data do not support the findings reported by the authors. In particular, the authors claim there is less nuclear β-catenin in the knockout cells, which does not appear to be easily observed in the images presented in Figure 6A.  Furthermore, they show an increase (albeit very small) in a β-catenin target gene Myc, which argues against a decrease in nuclear levels of β-catenin in Figure 6B.  As noted with the cell lines above, one gene is not sufficient to say that gene expression by a transcription factor is up or down regulated.  Additional, genes would need to be examined.    

      Response: Thanks the reviewer to highlighted these aspects. We performed new confocal microscopy experiments on SW480ctrl and SW480KCTD1-, shown in Figure 6 of the revised version of the manuscript. As can be seen, by also inserting a cytoskeleton dye such as Phalloidin, the increased nuclear localisation of beta-catenin in SW480KCTD1- cells compared to control cells is better observed. Furthermore, we added the following B-catenin target genes that de-regulate according to literature data (https://doi.org/10.1186/1471-2164-15-74): ABCB1, DKK4, ENFB1, LGR5, TCF1. These genes were also evaluated in CACO2 over expressing KCTD1 cells and the results were added in figure 7 of the revised manuscript version. The results section has been modified accordingly. 

  1. The follow-up study in CACO2 cells is interesting but does not fully address the concern I originally presented.  The same cell line should be used for both a knockdown and overexpression study.  Therefore, the authors need to show knockdown of KCTD1 in CACO2 has the opposite effect as overexpression or that overexpression in SW480 cells decreases cell growth. 

      Response: We performed an experiment of KCTD1 over-expression in SW480 cells for 24h. Since SW480 already possess very high levels of KCTD1, we limited ourselves to observing the effect of KCTD1 over-expression on B-Catenin levels. As shown in Supplementary Figure 2 of the revised manuscript, increasing KCTD1 levels leads to a significant reduction in B-catenin.

Overall, this study is interesting, but is missing key confirmatory experiments to show the importance of KCTD1 in colorectal cancer cell growth.

Reviewer 2 Report

Authors addressed some of my questions and fixed the typos I listed. I’m still concerning about the lacking of novelty. Careless mistakes in the manuscript are also concerning.

Major points:

1.      Authors clarified cell line was generated by CRISPR/CAS9, and revised as incomplete knock out, or knock down in the paper. A “clean” cell line propagated from a single cell is preferred for this kind of study

2.      Authors argued novel functions of KCTD1 in cell cycle, cell migration and possibly in colon cancer. Previous study also used colon adenocarcinoma cell line SW480, and also demonstrated KCTD1 regulating β‑catenin (DOI: 10.3892/mmr.2020.11457). β‑catenin is a well-known cell cycle regulator and promotes cell migration, indicating KCTD1’s role in the processes of cell cycle and cell migration. The major discoveries overlap with previous studies. Authors didn’t provide convincing evidence to support the necessity and significance of this research.

3.      The new data of KCTD12 and KCTD15 expression is nice and supports authors’ hypothesis.

Typos and careless mistakes in the previous version were fixed, but again there are new typos and careless mistakes in the updated supplementary figures. All the 4 new supplementary figures need improvements. Supplementary figure 1 has dashed lines on top of each panel. “2.55%” not aligned. Brightfiled images in supplementary figure 3 seem corresponding to figure 6, not figure 2. In supplementary figure 4, typos such as “analises”, “β‑catening”. In supplementary figure 6, “gray plot(s)”. “mRNA expression levels of KTCD proteins” is confusing. These mistakes won’t affect the conclusions of the study, but there are too many across the whole manuscript, which is really concerning to me.

Author Response

Reviewer 2

English language and style

( ) English very difficult to understand/incomprehensible
(x) Extensive editing of English language and style required
( ) Moderate English changes required
( ) English language and style are fine/minor spell check required
( ) I don't feel qualified to judge about the English language and style

Yes

Can be improved

Must be improved

Not applicable

Does the introduction provide sufficient background and include all relevant references?

( )

(x)

( )

( )

Are all the cited references relevant to the research?

(x)

( )

( )

( )

Is the research design appropriate?

(x)

( )

( )

( )

Are the methods adequately described?

(x)

( )

( )

( )

Are the results clearly presented?

( )

(x)

( )

( )

Are the conclusions supported by the results?

( )

(x)

( )

( )

Comments and Suggestions for Authors

Authors addressed some of my questions and fixed the typos I listed. I’m still concerning about the lacking of novelty. Careless mistakes in the manuscript are also concerning.

Major points:

  1. Authors clarified cell line was generated by CRISPR/CAS9, and revised as incomplete knock out, or knock down in the paper. A “clean” cell line propagated from a single cell is preferred for this kind of study.

Response: We fully understand the reviewer comment. Future studies will focus on the study of a cell line derived from a single silenced clone.

  1. Authors argued novel functions of KCTD1 in cell cycle, cell migration and possibly in colon cancer. Previous study also used colon adenocarcinoma cell line SW480, and also demonstrated KCTD1 regulating β‑catenin (DOI: 10.3892/mmr.2020.11457). β‑catenin is a well-known cell cycle regulator and promotes cell migration, indicating KCTD1’s role in the processes of cell cycle and cell migration. The major discoveries overlap with previous studies. Authors didn’t provide convincing evidence to support the necessity and significance of this research.

Response: We added new experimental data on the deregulation of important beta-catenin targets because of altered KCTD1 expression levels. We also extended our study to another colon cancer cell line, the CACO2 cell line, in which KCTD1 expression levels are low. The increase in KCTD1 results in opposite effects to cells in which KCTD1 was down regulated. The new experiments are reported in the revised version of the manuscript.

  1. The new data of KCTD12and KCTD15 expression is nice and supports authors’ hypothesis.

Typos and careless mistakes in the previous version were fixed, but again there are new typos and careless mistakes in the updated supplementary figures. All the 4 new supplementary figures need improvements. Supplementary figure 1 has dashed lines on top of each panel. “2.55%” not aligned. Brightfiled images in supplementary figure 3 seem corresponding to figure 6, not figure 2. In supplementary figure 4, typos such as “analises”, “β‑catening”. In supplementary figure 6, “gray plot(s)”. “mRNA expression levels of KTCD proteins” is confusing. These mistakes won’t affect the conclusions of the study, but there are too many across the whole manuscript, which is really concerning to me.

Response: Thanks the reviewer to highlighted these mistakes. We modified the revised version of the manuscript accordingly.

Round 3

Reviewer 1 Report

While I appreciate that the authors might not have access to the lines specifically mentioned, other lines could have been substituted.  However, I will accept the fact that the authors might not have access to lines other than SW480 and CACO2 cells.  The use of only these 2 lines greatly diminishes the impact of this study, and the field is moving towards requiring multiple cell lines to show an impact.  

Reviewer 2 Report

Manuscript revised accordingly.